# Is Device Removal Necessary after Fixed-Angle Locking Plate Osteosynthesis of Proximal Humerus Fractures?

**DOI:** 10.3390/medicina58030382

**Published:** 2022-03-04

**Authors:** Beom-Soo Kim, Du-Han Kim, Jung-Hoon Choi, Byung-Chan Choi, Chul-Hyun Cho

**Affiliations:** 1Department of Orthopedic Surgery, Keimyung University Dongsan Hospital, Keimyung University School of Medicine, Daegu 42601, Korea; kbs090216@gmail.com (B.-S.K.); osmdkdh@gmail.com (D.-H.K.); bcchoikr@dsmc.or.kr (B.-C.C.); 2Department of Orthopedic Surgery, Bogang Hospital, 102 Wolbae-ro, Dalseo-gu, Daegu 42801, Korea; cjh0487@naver.com

**Keywords:** shoulder, proximal humerus fracture, locking plate, device removal

## Abstract

*Background and Objectives*: The aim of this study was to evaluate whether device removal in symptomatic patients following locking plate osteosynthesis of a proximal humerus fracture improves the clinical outcomes. *Materials and Methods*: Seventy-one patients who underwent fixed-angle locking plate osteosynthesis of a proximal humerus fracture were included. Thirty-three patients underwent device removal at a mean time of 10.4 months after index surgery (removal group). Thirty-eight patients who retained the device after index surgery (retention group) were included in the control group. Visual analog scale (VAS) pain score, University of California at Los Angeles (UCLA) score, American Shoulder and Elbow Surgeons (ASES) score, and range of motion (ROM) were evaluated pre- and postoperatively. *Results*: At the final follow-up, mean UCLA score, ASES score, and all ROMs were significantly higher in the removal group compared to the retention group (*p* < 0.001). However, no significant difference in mean VAS pain score was observed between the two groups. Comparison of the clinical outcomes before and after device removal surgery showed significant improvement in all clinical scores and ROMs after device removal (*p* < 0.001). *Conclusions*: Device removal surgery in symptomatic patients following locking plate osteosynthesis of a proximal humerus fracture can result in significant improvement in functional outcomes.

## 1. Introduction

As we become an ageing society, the number of proximal humerus fractures is increasing [1]. Proximal humerus fractures may be a significant cause of morbidity with loss of independence [1,2]. Despite controversy regarding proper treatment of proximal humeral fractures, most surgeons support operative treatment of unstable or displaced fractures [2,3,4,5,6,7,8]. Compared to conservative treatment, open reduction and internal fixation (ORIF) enables the early return of function and decreases the incidence of malunion or nonunion [9]. Fixed-angle locking plate osteosynthesis for unstable proximal humerus fractures has recently become a well-established treatment option, and satisfactory clinical and radiographic outcomes were reported [2,3,4,5,6,7,10].

Despite advances in plate osteosynthesis and surgical techniques, functional impairment can persist. Hirschmann et al. [11] reported that 10 (17.5%) out of 57 patients who underwent ORIF for proximal humerus fractures had an unsatisfactory outcome with considerable pain or restricted motions at the long-term follow-up evaluation. After plate fixation, postoperative clinical outcomes can be influenced by implant-related complications, such as pain or tissue irritation around the retained implant or impaired function [10,12,13,14,15]. In orthopedic practice, these implant-related problems are indications for implant removal after fracture union [16,17,18,19]. 

Therefore, it is not surprising that device removal surgery accounts for 5–15% of all operations in the orthopedic and trauma unit [19]. Numerous studies have reported improvement in pain and functional outcomes after device removal surgery in other body parts; however, most device explanations regard the lower extremity [16,17,18,19]. Although device removal is usually regarded as a simple procedure [12], studies to determine the pros and cons of device removal after ORIF with a locking plate of a proximal humerus fracture are rare [10,12,13]. A few studies have reported clinical outcomes after fixed-angle locking plate removal of a proximal humerus fracture [10,12,13,20]. There is no guideline or consensus regarding whether or when the plate should be removed. Therefore, the aim of this study was to evaluate clinical outcomes after device removal following locking plate osteosynthesis of a proximal humerus fracture. This study was conducted to investigate the hypothesis that device removal in symptomatic patients following locking plate osteosynthesis of a proximal humerus fracture would result in significant improvement in clinical outcomes.

## 2. Materials and Methods

### 2.1. Study Design and Participants

A total of 213 patients who underwent ORIF for displaced proximal humerus fractures in a single institution from May 2009 to July 2019 were retrospectively reviewed. The inclusion criterion was a displaced proximal humerus fracture managed with a fixed-angle locking plate (PHILOS^®^; Synthes, Oberdorf, Switzerland) using a deltopectoral approach. For comparative analysis, patients were divided into two groups: the device removal group and the retention group. Exclusion criteria were as follows: (1) follow-up period less than 12 months after ORIF in the retention group; (2) follow-up period less than 12 months after device removal surgery in the removal group; (3) revision surgeries due to complications such as fracture nonunion or fixation failure; and (4) conversion to arthroplasty. Pain or tissue irritation around the retained implant or stiffness after fracture union in symptomatic patients was an indication for device removal surgery. Finally, 71 patients were included in this study. Thirty-three patients underwent device removal surgery at the mean time of 10.4 months after index surgery (removal group). Thirty-eight patients retained the implant after index surgery (retention group) (Figure 1). The study was approved by our Institutional Review Board (IRB No. 2020-09-035). Informed consent was waived due to the retrospective nature of the study design.

### 2.2. Surgical Procedure

An experienced surgeon performed ORIF using a PHILOS plate with the patient in the supine position under general anesthesia. Fracture fragments were reduced and fixed with a PHILOS plate through the deltopectoral interval. Proximal locking screws were usually placed into the head, and distal screws were fixed. Under fluoroscopic guidance, final plate placement and the length of all screws were assessed by taking the shoulder through multiple planes of motion. Tension sutures placed through the cuff tendons were also secured to the plate. 

In the removal group, device removal surgery was performed at a mean of 10.4 ± 6.7 months after index surgery. The initial deltopectoral approach was used again in all patients. After exposure of the plate, suture materials were removed if tension sutures were used during the initial operation. All screws and the plate were removed. Extra-articular adhesiolysis was performed simultaneously with device removal surgery. When scar tissue was observed around the subacromial space, exploration and debridement were performed to free the rotator cuff. If a heavy scar connecting the deep deltoid fascia and plate was observed, complete debridement of this scar was performed. After device removal surgery, active and passive range of motion (ROM) exercises were initiated immediately without sling immobilization. 

### 2.3. Assessment of Clinical Outcomes

Clinical outcomes were assessed using the University of California at Los Angeles (UCLA) score, the American Shoulder and Elbow Surgeons (ASES) score, the Visual Analogue Scale (VAS) pain score, and passive ROMs in four directions—forward flexion, abduction, external rotation with the arm at the side, and internal rotation at the back—at the final follow-up. Clinical outcomes were assessed before and after device removal surgery in the removal group. In the retention group, clinical outcomes were assessed at the final follow-up. Demographic data including age, sex, involved side, injury mechanism, presence of diabetes mellitus, time from initial injury to surgery, Neer classification, type of operation, and operative time were obtained from medical records.

### 2.4. Statistical Analysis

Statistical analysis was performed using the SPSS statistical package (version 20.0; IBM, Armonk, NY, USA). Data are described using frequencies for categorical variables, mean ± standard deviation for normally distributed continuous data. In statistical analysis, the association of variables between the removal group and the retention group was assessed using chi-square and unpaired *t*-tests. The clinical scores and ROMs before and after device removal were assessed using the paired *t*-test. For statistical analysis of internal rotation, values were converted into contiguously numbered groups: T1 through T12 to 1 through 12; L1 through L5 to 13 through 17; sacrum to 18; and buttock to 19. *p* < 0.05 was considered statistically significant. 

## 3. Results

Baseline characteristics are listed in Table 1. No significant differences regarding involved side, injury mechanism, diabetes mellitus, time from initial injury to surgery, Neer classification, type of operation, and operative time were observed between the two groups (*p* > 0.05). Significant differences regarding age (55.2 years in the removal group vs. 71.6 years in the retention group; *p* < 0.001) and sex (12 men:21 women in the removal group vs. 5:33 in the retention group; *p* = 0.022) were observed between the two groups.

At the final follow-up, significantly higher mean UCLA and ASES scores were observed in the removal group compared to the retention group (32.6 vs. 28.7 and 92.5 vs. 80.3; *p* < 0.001). However, no significant difference in mean VAS pain score was observed between the two groups (0.8 vs. 1.2; *p* > 0.05). Mean ROMs in four directions—forward flexion, abduction, external rotation, and internal rotation—were significantly higher in the removal group at the finial follow-up compared to the retention group (159.7° vs. 139.6°, 151.1° vs. 124.7°, 62.4° vs. 48.4°, and 10.0 vs. 13.3; all *p* < 0.001) (Table 2). 

Comparing the clinical outcomes before and at a mean follow-up period of 29.5 months after device removal surgery, all clinical scores and ROMs showed significant improvement after device removal (*p* < 0.001) (Table 3). Mean VAS pain, UCLA, and ASES scores improved from 2.8, 23.2, and 62.6 to 0.8, 32.6, and 92.5, respectively (all *p* < 0.001). Mean ROMs in four directions—forward flexion, abduction, external rotation, and internal rotation—improved from 126.7°, 114.7°, 35.3°, and 14.1 to 159.7°, 151.1°, 62.4°, and 10.0, respectively (all *p* < 0.001). 

Among 33 cases of device removal surgery, a complication with a screw jamming occurred in a patient who underwent plate removal at 35.9 months after ORIF. The jammed screw was cut by a high-speed burr and the remaining plate was removed. There were no other complications, including iatrogenic fracture, neurovascular injury, infection, or instability. None of the patients underwent revision surgery, such as arthroscopic capsular release. 

## 4. Discussion

The findings of this study showed that device removal surgery in symptomatic patients following locking plate osteosynthesis of a proximal humerus fracture can result in significant improvement in functional outcomes without complications. Significantly higher functional scores and ROMs were observed in the removal group compared to the retention group at the final follow-up. All clinical scores and ROMs showed significant improvement after device removal in comparison to the preoperative status.

There are possible benefits of device removal surgery in patients after proximal humerus fractures are treated by ORIF with a locking plate. First, device removal surgery may potentially minimize the risk of glenoid destruction by screw cut-out [13]. Glenoid destruction by a locking screw is the most devastating, subsequently requiring arthroplasty [5,7,21]. Dimitriou et al. [13] reported that early plate removal at a minimum of 6 months after ORIF in radiographically consolidated fractures may potentially reduce the risk of implant-related complications, such as secondary screw cut-out due to avascular necrosis (AVN) of the humeral head. They emphasized that early plate removal may improve clinical outcomes in both relatively asymptomatic patients and symptomatic patients [13]. Second, the device removal procedure including simultaneous peri-implant adhesiolysis may help improve the clinical symptoms. Bhatia et al. [22] reported that severe adhesion in the subdeltoid area between the deltoid and the plate was observed in patients with persistent postsurgical stiffness following locking plate osteosynthesis for proximal humerus fractures. Holloway et al. [14] reported that complete debridement of heavy scars connecting the acromion and the deep deltoid fascia is important to regain maximal ROM. Implant retention in surrounding tissue can cause extraarticular adhesion, causing pain and limited ROM in the patient. In the current study, we performed device removal surgery in symptomatic patients as soon as possible if the fracture was united completely, which resulted in significant improvement in clinical outcomes. We think that extraarticular adhesiolysis performed simultaneously with the removal surgery might have been helpful in regaining ROM. 

Although device removal procedures are frequently regarded as a simple and common operation [12], studies on the characteristics of patients who underwent plate removal after ORIF for proximal humerus fractures are rare [10,12,13]. In the current study, significant differences in terms of age (55.2 years in the removal group vs. 71.6 years in the retention group) and sex (12 men:21 women in the removal group vs. 5:33 in the retention group) were observed between the two groups. However, regarding the involved side, injury mechanism, diabetes mellitus, time from initial injury to surgery, Neer classification, type of operation, and operative time, no significant differences were observed between the two groups. The results showed that elderly patients often had a comorbid disease and were worried about the risk of anesthesia and second surgery. Proximal humerus fractures in young patients tend to occur from high-energy trauma and are accompanied by more serious soft tissue damage which may affect postoperative clinical outcomes. In addition, younger patients, particularly active men, with functional deficits were inclined to want to remove the plate. 

The timing of plate removal surgery after ORIF using a locking plate for proximal humerus fractures is poorly understood. Acklin et al. [12] reported that plate removal was performed after a mean time of 13 ± 5 months. Kirchhoff et al. [10] reported that the indication for device removal was stated earliest at 12 months after index surgery. Dimitriou et al. [13] reported that early device removal (around 6 months after index surgery) in all patients was recommended because of secondary screw cut-out after humeral head AVN. Several studies reported that clinical scores and ROMs reach a plateau at approximately 6–12 months after ORIF for proximal humerus fractures [6,23]. In the current study, device removal surgery was performed at a mean time of 10.4 ± 6.7 months after index surgery. Considering these results, we think that device removal surgery after bone healing should be recommended as soon as possible for patients who have persistent pain and limited ROMs after sufficient recovery time. 

Only three studies have analyzed the outcomes of device removal surgery after locking plate fixation for proximal humerus fractures [10,12,13]. Kirchhoff et al. [10] reported that the mean Constant score showed significant improvement from 66.2 preoperatively to 84.3 after device removal without complications. Acklin et al. [12] reported the mean Constant score showed significant improvement from 71 to 76. Dimitriou et al. [13] reported that the mean Constant score and subjective shoulder value were 83.8 and 92.8 at 12 months after device removal, respectively. However, these studies did not include a comparison of clinical outcomes between patients who underwent device removal surgery and those who retained the implant without removal surgery. In the current study, mean UCLA score (32.6 vs. 28.7), ASES score (92.5 vs. 80.3), and all ROMs (forward flexion, 159.7° vs. 139.6°; abduction, 151.1° vs. 124.7°; external rotation, 62.4° vs. 48.4°; internal rotation, 10.0 vs. 13.3) were significantly higher in the removal group than in the retention group at the final follow-up. A comparison of the clinical outcomes before and after device removal surgery showed significant improvement in mean VAS pain, UCLA, and ASES scores from 2.8, 23.2, and 62.6 to 0.8, 32.6, and 92.5, respectively. Mean ROMs in four directions—forward flexion, abduction, external rotation, and internal rotation—showed significant improvement from 126.7°, 114.7°, 35.3°, and 14.1 to 159.7°, 151.1°, 62.4°, and 10.0, respectively.

Katthagen et al. [23] reported significant improvement in functional outcomes and patient satisfaction after arthroscopic surgery in symptomatic patients after locking plate fixation for proximal humerus fractures. They emphasized that the capsular release in case of sequelae of plating for proximal humerus fractures is a crucial aspect of arthroscopy, which cannot be carried out to the same extent during open surgery [23]. In contrast to the opinions described by Katthagen et al. [23], none of the patients in the current study underwent arthroscopic revision surgery such as capsular release. Nonetheless, clinical scores and ROMs showed significant improvement in all patients after device removal. We think that the functional impairment following ORIF for proximal humerus fractures might be more related to extraarticular adhesion. Therefore, in symptomatic patients, device removal surgery including the adhesiolysis of extraarticular adhesion was considered and sufficient to improve clinical outcomes. 

There are several limitations to the study. First, this study had a retrospective design with a small sample size. Second, there was a potential bias due to a substantial loss of follow-up evaluations. Third, there was no evaluation of accompanying injuries, such as rotator cuff tears, which might be considered potential sources of pain or functional deficits. Fourth, there were many patients who did not undergo device removal surgery due to the risk of general anesthesia in older patients, so the clinical results were poor in the retention group. Well-designed, prospective, randomized, controlled trials are needed to develop clear guidelines for device removal surgery. 

## 5. Conclusions

Device removal surgery in symptomatic patients following locking plate osteosynthesis of a proximal humerus fracture can result in significant improvement in functional outcomes.

## Figures and Tables

**Figure 1 medicina-58-00382-f001:**
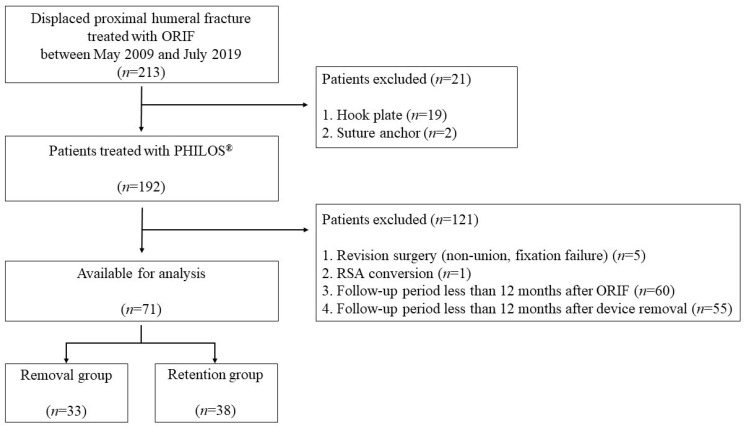
Flowchart of patient inclusion. ORIF: Open Reduction and Internal Fixation.

**Table 1 medicina-58-00382-t001:** Demographic factors of device removal and retention groups.

	Removal Group (*n* = 33)	Retention Group (*n* = 38)	*p* Value
Age, year	55.2 ± 12.0	71.6 ± 9.5	<0.001 *
Sex, male/female, n	12/21	5/33	0.022 *
Involved side, right/left, n	19/14	26/12	0.344
Injury mechanism			0.053
Low-energy trauma	18	12	
High-energy trauma	15	29	
Diabetes mellitus, yes/no, n	5/28	13/25	0.066
Time from initial injury to surgery, day	4.8 ± 3.5	5.4 ± 4.8	0.567
Fracture Classification (Neer)			0.619
2-part	21	19	
3-part	9	18	
4-part	3	1	
Type of operation			0.068
Plate + tension suture	25	22	
Plate + tension suture + allograft	3	12	
Plate only	5	4	
Operative time (ORIF), minute	98.1 ± 33.3	102.5 ± 45.2	0.650
Time after ORIF to implant removal, month	10.4 ± 6.7		
Total follow-up periods, month	39.9 ± 20.3	22.4 ± 14.9	<0.001 *

ORIF, open reduction and internal fixation; Low-energy trauma, slip down; High-energy trauma, motor vehicle accident or fall from height; values are presented as mean ± standard deviation; * statistically significant, *p* < 0.05.

**Table 2 medicina-58-00382-t002:** Comparison of clinical outcomes between device removal and retention groups.

	Removal Group (*n* = 33)	Retention Group (*n* = 38)	*p*-Value
Finial Clinical scores			
VAS pain score	0.8 ± 1.5	1.2 ± 1.1	0.283
UCLA score	32.6 ± 3.2	28.7 ± 3.7	<0.001 *
ASES score	92.5 ± 9.8	80.3 ± 10.6	<0.001*
Final Range of motion			
Forward flexion, °	159.7° ± 18.5°	139.6° ± 22.9°	<0.001 *
Abduction, °	151.1° ± 30.7°	124.7° ± 24.7°	<0.001 *
External rotation, °	62.4° ± 13.2°	48.4° ± 14.8°	<0.001 *
Internal rotation	10.0 ± 2.4	13.3 ± 2.6	<0.001 *

UCLA, University of California at Los Angeles; ASES, American Shoulder and Elbow Surgeons; VAS, visual analog scale; values are presented as mean ± standard deviation; * statistically significant, *p* < 0.05.

**Table 3 medicina-58-00382-t003:** Comparison of clinical outcomes before and after implant removal surgery.

	Preoperative	Postoperative	*p*-Value
Clinical scores			
VAS pain score	2.8 ± 2.0	0.8 ± 1.5	<0.001 *
UCLA score	23.2 ± 6.0	32.6 ± 3.2	<0.001 *
ASES score	62.6 ± 16.7	92.5 ± 9.4	<0.001 *
Range of motion			<0.001 *
Forward flexion, °	126.7° ± 31.0°	159.7° ± 18.5°	<0.001 *
Abduction, °	114.7° ± 31.6°	151.1° ± 30.7°	<0.001 *
External rotation, °	35.3° ± 17.5°	62.4° ± 13.2°	<0.001 *
Internal rotation	14.1 ± 3.3	10.0 ± 2.4	<0.001 *

UCLA, University of California at Los Angeles; ASES, American Shoulder and Elbow Surgeons; VAS, visual analog scale; values are presented as mean ± standard deviation; * statistically significant, *p* < 0.05.

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
