# Peer review of "Is Device Removal Necessary after Fixed-Angle Locking Plate Osteosynthesis of Proximal Humerus Fractures?"

_medicina, 2022, doi:10.3390/medicina58030382_

Round 1

Reviewer 1 Report

Manuscript titled, "Is device removal necessary after fixed-angle locking plate osteosynthesis of proximal humerus fractures?", evaluated whether device removal in symptomatic patients following locking plate osteosynthesis of the proximal humerus fracture improves the clinical outcomes. Results of this study showed that Device removal surgery in symptomatic patients following locking plate osteosynthesis of a proximal humerus fracture can result in a significant improvement of functional outcomes. The manuscript is well written and the conclusion reported in the study is well supported by the results presented in the study.

Author Response

Thanks for your kind comment.

Reviewer 2 Report

This paper is interesting for the timing and necessity of implant removal in patients with persistent pain and loss of motion after proximal humerus fracture.

There are several minor problems with the study.

First, there is a significant age difference between two groups, does it cause lower ROM and clinical outcome in the retention group?

Second, when did you measure the clinical outcome in retention group and removal group?  

Author Response

First, there is a significant age difference between two groups, does it cause lower ROM and clinical outcome in the retention group?

-> Thanks for your kind comment. There was no bias in collecting and analyzing the data for this study, and these results were obtained by organizing the data as it is. There was difference of age in two group, we know. Due to the nature of this fracture, it frequently occurs in the elderly, and instrument removal was not usually done. And, due to the cultural characteristics of the country where the study was conducted, there are cases where the elderly does not want surgery because of the risk of general anesthesia even if they have pain or joint symptoms, and young patients tend to prefer instrument removal. That was also our limitation of this study. We added that limitation in the Discussion part.

Second, when did you measure the clinical outcome in retention group and removal group?

->Thanks for your kind comment. In removal group, we measured at final follow-up period. We added that in Method part.
